# An Interaction between Brain-Derived Neurotrophic Factor and Stress-Related Glucocorticoids in the Pathophysiology of Alzheimer’s Disease

**DOI:** 10.3390/ijms25031596

**Published:** 2024-01-27

**Authors:** Tadahiro Numakawa, Ryutaro Kajihara

**Affiliations:** 1Department of Cell Modulation, Institute of Molecular Embryology and Genetics, Kumamoto University, Kumamoto 860-0811, Japan; 2Department of Biomedical Laboratory Sciences, Faculty of Life Science, Kumamoto University, Kumamoto 862-0976, Japan

**Keywords:** brain-derived neurotrophic factor, TrkB, intracellular signaling, synaptic plasticity, glucocorticoids, GR, depression, Alzheimer’s disease

## Abstract

Both the brain-derived neurotrophic factor (BDNF) and glucocorticoids (GCs) play multiple roles in various aspects of neurons, including cell survival and synaptic function. BDNF and its receptor TrkB are extensively expressed in neurons of the central nervous system (CNS), and the contribution of the BDNF/TrkB system to neuronal function is evident; thus, its downregulation has been considered to be involved in the pathogenesis of Alzheimer’s disease (AD). GCs, stress-related molecules, and glucocorticoid receptors (GRs) are also considered to be associated with AD in addition to mental disorders such as depression. Importantly, a growing body of evidence suggests a close relationship between BDNF/TrkB-mediated signaling and the GCs/GR system in the CNS. Here, we introduce the current studies on the interaction between the neurotrophic system and stress in CNS neurons and discuss their involvement in the pathophysiology of AD.

## 1. Introduction

In the mammalian brain, the brain-derived neurotrophic factor, BDNF, which is the most extensively studied neurotrophins, has been recognized as a critical player in promoting neuronal survival and differentiation as well as regulating synaptic plasticity. A growing body of evidence indicates that the protective effects of BDNF against neural damage in the central nervous system (CNS) occur by activating TrkB, a high affinity receptor for BDNF, although its precursor molecule, proBDNF, binding to p75NTR, which is the first identified common receptor for neurotrophins, including the nerve growth factor (NGF), neurotrophin-3 (NT-3), and neurotrophin-4 (NT-4), also negatively affects neuronal aspects such as neuronal survival and functions [1,2,3]. The critical involvement of BDNF/TrkB-mediated signaling, such as phospholipase Cγ (PLCγ)-, PI3k/Akt-, and ERK-signaling in the neuro- and glio-genesis, as well as in synaptic function, is highlighted [4]. As expected, in addition to experimental research studies, clinical evidence also shows that the alteration in BDNF levels in neurodegenerative diseases can be a promising biomarker in most neurodegenerative conditions and neurodegenerative diseases, including Alzheimer’s disease (AD) [5]. Furthermore, it has been demonstrated that apoptotic elements, which are considered responsible for manifestations linked to the pathophysiology of AD, interact with various signaling molecules, including BDNF/TrkB, and downstream signaling pathways [6]. Also, clinical and preclinical research demonstrates that the alteration of BDNF/TrkB-mediated signaling is involved in the pathology of depression [7].

In order to cope with a broad spectrum of stressful stimuli, the hypothalamic–pituitary–adrenal (HPA) axis functions as a critical neuroendocrine system. The regulation of blood concentrations of glucocorticoids (GCs), which are secreted from the adrenal glands on top of the kidneys, is achieved through the negative feedback mechanism of the HPA axis, while abnormally increased GC levels may be induced under a chronic stressful condition and cause dysfunction of the brain. Importantly, it has been known that uncontrollable stress influences the hippocampus at various levels, for example, hippocampus-dependent memory tasks and synaptic plasticity [8]. Structurally, studies have shown that stress changes neuronal morphology and reduces neuronal proliferation and hippocampal volume [8]. Such dysregulation of the HPA axis has been suggested to be involved in the pathogenesis of not only mental disorders but also AD [9]. As expected, a variety of studies show that GCs affect neurons and induce neurobiological changes (molecular and cellular levels) via activation of their receptors, the glucocorticoid receptor (GR), or the mineralocorticoid receptor (MR). Therefore, studies on the influence of GCs on neuronal aspects such as cell proliferation and cell survival, neurogenesis, synaptic function, genetic vulnerability, and epigenetic alterations are extensively performed to clarify the basis of brain diseases [10]. In this review, we introduce current studies concerning the possible interaction of the BDNF/TrkB system with GCs/receptor functions in the CNS neurons. Furthermore, we discuss the involvement of the interplay of BDNF and GCs in the pathophysiology of AD.

## 2. BDNF/TrkB System in Neuronal Function

BDNF is a member of the neurotrophin family, which consists of NGF, NT-3, and NT-4, and is most widely distributed in the mammalian brain, playing multiple roles in synaptic plasticity. Each neurotrophin has a high-affinity receptor. NGF binds to TrkA with high affinity. Similarly, both BDNF and NT-4 bind to TrkB, and NT-3 also binds to TrkC with high affinity. As a common receptor, p75NTR also associates with all neurotrophins and contributes to neuronal responses such as cell death, although the Trks have been considered to be involved in a variety of positive effects, including cell survival and synaptic regulation [11]. Importantly, when BDNF binds to the high-affinity receptor, mainly TrkB, three signaling pathways, including PLCγ, PI3k/Akt-, and ERK-signaling, are activated to affect neuronal events such as synaptic transmission (see Figure 1). Because the BDNF/TrkB system has critical roles in the various brain regions, including hippocampal and cortical areas, which are required for learning and memory functions, it is demonstrated that the downregulation of the BDNF/TrkB system has a link to the pathogenesis of various brain diseases, including AD and mental disorders. In the embryonic and adult stages, an important contribution of BDNF/TrkB to neurogenesis is well recognized, and the changed neurogenesis is also associated with the pathophysiology of these brain diseases [12]. It is well known that these TrkB-dependent signaling pathways and resultant neural cell events are stimulated by mature (processed) BDNF. Firstly, the BDNF molecule is translated as a precursor protein, proBDNF, and it is further cleaved into the small mature molecule that binds to the TrkB receptor with high affinity. In contrast, p75NTR functions as a high-affinity receptor for proBDNF while having low affinity for mature neurotrophins, including mature BDNF, and the p75NTR-mediated signaling exerts a negative impact on neurons, including cell death induction and decreased synaptic function [1,2,3].

The role of the TrkB.T1 isoform, an alternative splicing of TrkB, is also intensively studied in this study because of its biological contribution to the CNS [13]. TrkB.T1 isoform has a truncated C-terminal domain and lacks the intracellular domain of the full-length receptor TrkB (TrkB.FL), resulting in the loss of autophosphorylation activity, which is required to trigger intracellular signaling. Recently, the possible involvement of an imbalance between these TrkB isoforms in aberrant signaling and hyperpathia pain has been demonstrated [13]. The evidence suggests that the truncated receptor, TrkB.T1, is harmful when it is upregulated in response to an injury or inflammation, although the BDNF/TrkB (the FL type) system is essential for brain development and maintenance in adulthood, including contributing to cell survival promotion and the positive regulation of synaptic function [13]. It is possible that various neuronal responses depend on the expression balance of pro/mature BDNF, various TrkB receptor isoforms, and p75NTR in the brain regions. 

It was reported that the transplantation of neural stem cells (NSCs) after hypoxic treatment increased the neuronal survival of ChAT (choline acetyltransferase)-positive neurons in a spinal cord injury (SCI) rat model. In the system, the upregulation of growth factors including NT-3, glial cell-derived neurotrophic factor (GDNF), and BDNF was observed, suggesting that the transplantation of NSCs with hypoxic preconditioning to increase these trophic factors is an effective strategy for cell-based therapy in the treatment of SCI [14]. Interestingly, a recent study has demonstrated the promotion of axonal remodeling and restoration of abnormal synaptic structures using adeno-associated virus (AAV) vectors carrying genes encoding BDNF or TrkB in the stroke model [15]. After the middle cerebral artery occlusion (MCAO), the stroke rats received a combination therapy with AAV-BDNF and AAV-TrkB, and they showed significantly improved upper-limb motor functional recovery and neurotransmission efficiency compared to the AAV vector treatment alone. It was also revealed that increased levels of synapsin I, postsynaptic density protein 95 (PSD-95), and GAP-43 were achieved by the combination therapy, strongly suggesting the importance of the BDNF/TrkB system in the functional recovery of an injured CNS [15]. 

The glial population is also affected by BDNF. Astrocytes have a role in neuronal survival and differentiation, and they support oligodendrocytes to maintain brain function. A recent report on astroglial functions has shown that the morphology, physiology, and survival of both oligodendrocytes and neurons were affected by the astrocytic Methyl-CpG-Binding Protein 2 (MeCP2). Interestingly, the BDNF mRNA expressions and secreted BDNF protein levels in astrocytes change after astroglial MeCP2 knockdown [16]. Datta et al. (2018) showed significant differences in dopaminergic neuronal cell survival co-cultured with midbrain astrocytes under 6-OHDA stress in comparison to hindbrain and forebrain astrocytes [17]. They also found that an increased secretion of BDNF in midbrain astrocytes was noted compared with hindbrain and forebrain astrocytes in the presence of 6-OHDA, suggesting that astrocytic BDNF contributes to the survival of dopaminergic neurons [17]. It was revealed that BDNF prevented astroglial cells from apoptosis via TrkB-T1-dependent intracellular signaling [18]. Interestingly, such an anti-apoptotic action of BDNF was abolished in the presence of a TrkB antagonist. The activation of signaling molecules including ERK, Akt, and Src (a non-receptor tyrosine kinase) by BDNF was also observed, although astrocytes only express TrkB-T1, suggesting the contribution of TrkB-T1 in intracellular signaling for astroglial protection [18]. Furthermore, an upregulation of TrkB.T1 has been suggested to play a role in the pathogenesis of various neuropathic disorders [13]. A growing body of evidence suggests roles of elevated TrkB.T1 in astrocytes and post-injury reactive astrogliosis in multiple CNS injury models [13]. Surprisingly, Harley et al. (2021) reported that the selective ablation of BDNF in microglia, which are resident immune cells in the CNS, enhanced the production of newborn neurons under inflammatory conditions in which LPS (lipopolysaccharide)-induced infections or traumatic injuries of the brain were conducted [19]. It was revealed that BDNF ablation in microglia interfered with self-renewal/proliferation and caused a decrease in their overall density, implying an influence of the microglial BDNF on neurogenesis via regulating microglia population dynamics and states [19]. 

In addition, cerebral endothelial cells (CECs), forming the blood –brain barrier (BBB), are also a potential source for bioactive BDNF and crucial regulators of brain homeostasis [20]. It is reported that endothelial BDNF synthesis/secretion is mainly controlled by nitric oxide (NO), the best-characterized endothelium-derived factor [21]. Several studies have demonstrated the physiological roles of cerebral endothelial BDNF, such as neurogenesis, neuroprotection, cerebral angiogenesis, and neuroplasticity [21]. It is of importance that cerebral endothelial BDNF expression is reduced in animal models of hypertension, diabetes, and rheumatoid arthritis, indicating that diminished endothelial BDNF expression could be a novel marker of endothelial dysfunction [21]. Various BDNF actions not only in neuronal and glial cells but also in cerebral microvasculature should be further studied in a variety of brain regions.

## 3. Glucocorticoids and Neuronal Functions

It is well known that the function of the HPA axis, which is critically important to cope with stressful conditions and strictly regulates blood levels of GCs, has been extensively studied, as dysregulation of the HPA axis has been suggested to be one of the risk factors in the pathogenesis of mental disorders and AD [9]. Importantly, secreted GCs from the adrenal glands, located on top of the kidneys, contribute to regulated blood levels of GCs via the negative feedback mechanism of the HPA axis. Since the HPA axis is considered a key endocrine system that has a critical role in the coordination of body-wide changes to survive stress, a lot of studies are focusing on the relationship between depression and the dysfunction of the HPA axis [22]. Recent evidence also shows changed neuronal function of corticotrophin-releasing hormone (CRH) neurons in the paraventricular nucleus of the hypothalamus (PVN) is involved in stress-related behaviors [22]. Generally, an activation of the HPA axis, which is required for the secretion of GCs acting on a variety of organs to cope with stress during the process of adaptation, invokes CRH release from PVN neurons. 

The roles of GCs and their receptors, including the glucocorticoids receptor (GR, low affinity) and mineralocorticoid receptor (MR, high affinity), in the CNS neurons have been discussed since several neurobiological changes (genetic, epigenetic, molecular, and cellular levels) caused by released GCs/receptors under chronic stress are closely associated with negative consequences, including psychiatric disorders. Therefore, in vitro studies examining the impacts of GCs on neuronal aspects such as cell proliferation and survival, neurogenesis, neurotransmission, genetic vulnerability, epigenetic alterations, and inflammation are very important [10]. Therefore, in this section, current studies concerning the contributions of GCs/receptors to neuronal aspects have been introduced. 

Recently, Wei et al. (2023) reported a high emotional reactivity of forebrain GR overexpression mice [23]. They showed a prolonged duration of anxiety behavior and upregulation of cFos co-labeling in the calbindin1 + glutamatergic neurons in the ventral hippocampus CA1 and in the DARPP-32 (dopamine- and cAMP-regulated phosphoprotein of M(r) 32 kDa) + glutamatergic neurons in the posterior basolateral amygdala after an optogenetic stimulation in the ventral dentate gyrus of GR overexpression mice [23]. A study by Dwyer et al. (2023) showed that a novel neurosteroid, NTS-105, has beneficial effects against traumatic brain injury (TBI). Using organotypic hippocampal slice cultures, it was revealed that deficits in the hippocampal LTP (long-term potentiation, an important form of synaptic plasticity) were prevented by post-traumatic administration of NTS-105, and post-traumatic cell death was also decreased by NTS-105 treatment [24]. Interestingly, the novel neurosteroid inhibited the activation of the MR and the androgen receptor, while it was not active at the GR [24]. We recently performed a social reward-conditioned place preference (SCPP) test using AKR-, BALB/c-, and C57BL/6J-strain mail mice and found significant anxiety-like and anhedonia-like behaviors in BALB/c mice with downregulation of GR in the nucleus accumbens (NAC), compared with those in AKR and C57BL/6J strains [25]. We previously reported that GC exposure caused a marked downregulation of GR in cultured cortical and hippocampal neurons and that the GR downregulation negatively affected synaptic functions, including the release of glutamate while MR expression was unchanged by GC exposure [26,27], suggesting possible involvement of GR (not MR) in synaptic regulation. Recently, McCann et al. (2021) reported the highest density of MRs and MR:GR ratio in the hippocampal CA2 area compared with all other subregions, including CA1, CA3, and the dentate gyrus (DG), in adult mice [28]. They also found that the MR in the CA2 area is required for the acquisition of area-specific genes and protein expression in pyramidal neurons, suggesting important roles for the MR in the hippocampal area [28].

It is recognized that females have higher incidences of affective disorders, including anxiety, PTSD, and major depression, although mechanisms under the sex biases remain unclear. Montgomery et al. (2023) showed an acute activation of the HPA axis by clozapine-N-oxide (CNO) in the transgenic mice expressing the Gq-coupled Designer Receptor exclusively stimulated by the Designer Drugs (DREADD) receptor hM3Dq in corticotropin-releasing factor (CRF, or CRH) neurons and found a novel sex-specific dissociation of GCs levels using the CRH activation method [29]. Although hM3Dq-expressing males displayed physiological stress sensitivity, including reductions in body and thymus weights, the corticosterone levels in response to CNO in the hM3Dq females were greater than those in males. The hM3Dq female animals also showed significant increases in baseline and fear-conditioned freezing behaviors, suggesting a possible contribution of activation of CRH neurons to sex-specific behaviors [29]. You et al. (2023) used a predator scent stimulus (PSS) paradigm where mice were exposed to a volatile predator cue (e.g., cat odor) as a psychological stressor and found that appetite changes were caused by PSS [30]. Interestingly, in their system, proenkephalin (Penk)-expressing lateral hypothalamic (LHPenk) neurons contributed to overconsumption of a high-fat diet (HFD) because silencing the LHPenk neurons normalizes the HFD overconsumption caused by PSS. They also found that elevated corticosterone has a role in the reactivity of GR-containing LHPenk neurons to HFD, as inhibition of the GR resulted in the suppression of PSS-induced responses, demonstrating a critical role of physiological/hormonal alterations in appetite regulation via affecting feeding-related hypothalamic circuit functions [30].

As mentioned above, GCs, the principal effectors of stress, are associated with multiple signaling in neurons due to a variety of membrane and nuclear receptor subtypes [31]. It has been suggested that both GR and MR for GCs are localized to the cell membrane, cytosol, and nucleus, resulting in multiple intracellular signaling and in different time scales of regulation of synaptic function. Interestingly, the growing evidence also demonstrates that the rapid actions of acute stress-induced GCs, including changes in synaptic transmission and neuronal excitability, depend on the activation of membrane GRs and MRs [31].

Markedly, a growing body of evidence demonstrates that impaired mitochondrial function is associated with major depressive disorder, and a relationship between mitochondrial dysfunction and GC stress is also important to understand the pathogenesis of depressive disorder [32]. Indeed, studies have demonstrated the critical influence of the mitochondrial system on cognitive processes in the mature brain via regulating NSC fate during neurodevelopment [33] and the role of mitochondrial function in the NAc and CRH neurons in anxiety and stress [34]. Collectively, the influence of the HPA axis system, including CRH and GCs, in the CNS neurons is evident; therefore, understanding how the altered function of both mitochondrial and GC stress systems affects the BDNF/TrkB system is very interesting.

## 4. Glucocorticoid Stress, BDNF, and Neuronal Functions

The evidence suggests an involvement of altered blood concentrations of GCs in the pathogenesis of various brain diseases in humans. In addition, in adult mammals and humans, it has been demonstrated that neuroplasticity, which is controlled through various mechanisms, is influenced by GC stress. As expected, regulated levels of blood GCs are closely associated with brain function by affecting hippocampal plasticity, including glutamatergic neurotransmission, neurogenesis, systems of neurotrophic factors, microglia, astrocytes, neuroinflammation, and so on [35]. In addition to the amygdala and hippocampus, the prefrontal cortices are involved in the corticolimbic system and are perturbed by the abnormal function of BDNF/TrkB-related signaling and activity of GR [36]. The long-term effects of developmental stressors on the BDNF/TrkB system and GR activity in the prefrontal cortices have also been argued to understand the possible pathogenesis of neuropsychiatric illnesses, including depression [36]. For example, a recent study has demonstrated the involvement of BDNF-mediated GR function in long-term memory retention. Arango-Lievano et al. (2019) found that either the lack of GR-phosphorylation (PO4), which was caused by mutating the BDNF-dependent PO4 sites, or decreased secretion of BDNF using the BDNF-Val66Met mutant resulted in the reduced maintenance of newly formed postsynaptic dendritic spines in the cortex and caused a malfunction in the memory retention of mice [37]. The interplay between the BDNF/TrkB system and the HPA axis plays a central role in the healthy functioning of the brain, and it is probably that the changed interaction of BDNF/TrkB signaling with GR activity is involved in brain dysfunction. 

As a useful experimental strategy, excess GC exposure using animals has been extensively performed. Using postnatal Dairy Bull Calves, an examination of leptin concentrations and brain development markers, including BDNF, after the exogenous GC administration was carried out. McCarty et al. (2023) showed a decreased expression of BDNF, FGF1 (fibroblast growth factor), and FGF2 in the hypothalamic of Holstein bulls at 5 days after the intravenous infusion of cortisol within 4 h of parturition, followed by a second infusion 24 h postpartum, with reduced leptin levels in the blood and cerebrospinal fluid [38]. DEX, a steroid medicine, is used to treat a number of conditions, including inflammatory disorders; thus, it is also used to examine the impact of early-life DEX application on hippocampal BDNF expression. Chen et al. (2020) performed a daily intraperitoneal injection of DEX in mice from postnatal 1 (P1) to P7 and examined the hippocampal BDNF levels in both the perinatal period and adulthood [39]. Although the neonatal DEX (ND) treatment led to depressive-like behaviors in adulthood (P90–P110), the expression of hippocampal BDNF was unchanged. However, after ND exposure, downregulated BDNF levels in the perinatal period and puberty (P40) were observed, suggesting that early-life DEX exposure caused interference in BDNF signaling during neuronal development [39]. A lot of evidence has shown the downregulation of the BDNF system after GC stress [9] (see Figure 1).

Using mice overexpressing TrkB.T1, Razzoli et al. (2011) examined behavioral phenotypes after repeated social defeat stress and found that the TrkB.T1 mice, which showed significant downregulation of the hippocampal BDNF protein, displayed more consistent social avoidance effects than their wild-type littermates [40]. It was reported that the BDNF/TrkB system and synaptic plasticity in the amygdala were involved in an age-dependent effect induced by stress on the fear-potentiated memory [41]. After subjecting adolescent (4-week-old) and adult (8-week-old) rats to social instability stress for 5 weeks, it was revealed that the stress appeared to hinder the fear-potentiated startle responses in adolescents, although it improved those in adult animals. Interestingly, stress in adolescence reduced the expression of synaptic proteins, including full-length TrkB and SNAP-25 in the amygdala, whereas adult animals treated with stress showed increased amygdala expression of full-length and truncated TrkB expressions [41]. Moreover, the effects of adolescent corticosteroid exposure on the balance between full-length and truncated TrkB were shown [42]. Barfield et al. (2017) reported that mice that received CORT exposure exhibited a decreased ratio of full-length TrkB/TrkB.T1 in the medial prefrontal cortex, ventral hippocampus, and amygdala, with decreased hippocampal ERK-signaling [42]. Interestingly, using 10-day combinatory stress paradigms (postnatal days 26 to 35), Azogu et al. (2018) found sex-specific differences in both endocrine and plasticity-signaling responses in adulthood when animals received repeated stress during their adolescent period [43]. They reported elevated levels of basal and stress-induced CORT in females compared with males, which were attenuated by ANA-12, a small-molecule TrkB antagonist, post-stress in both sexes. As adults, females displayed higher exploratory and locomotor activity than males. In non-stress animals, males showed elevated TrkB.T1 and full-length TrkB compared to females in the PFC, hippocampus, and NAc in vehicle-treated animals. On the other hand, stress exposure caused significant downregulation of both TrkB.T1 and TrkB.FL in the NAc of males only, and it was associated with upregulation of TrkB.T1 in the PFC [43]. Recently, NAc cell subtype contributions of the BDNF/TrkB system in the stress outcomes have been demonstrated [44]. TrkB.T1 overexpression in the NAc dopamine receptor 2 (D2)-containing medium spiny neurons (MSNs) prevents chronic social stress outcomes in mice, although this overexpression in NAc D1-MSNs potentiated stress-susceptible outcomes after the social defeat stress paradigm [44]. Increasing evidence suggests that the TrkB.T1 function, including Ca^2+^ release from the intracellular storage instead of lacking studies on molecules interacting with the truncated receptor, is also involved in BDNF signaling in both normal physiological and pathological conditions [45].

Using AKR-, BALB/c-, and C57BL/6J-strain male mice, we examined depression-like behaviors, social reward responses, and BDNF expression [25]. Interestingly, among them, the BALB/c line exhibited the highest anxiety-like and anhedonia-like behaviors, and also displayed increased responses to social rewards in SCPP, with downregulated GR, and no changed protein levels of MR, TrkB, and p75 in the NAC, suggesting possible involvement of NAC GR activity in an increased social reward response and anxiety [25]. In contrast, after preadolescent rats were exposed to an enriched environment (EE) and combined exercise training (CET), it was revealed that hippocampal GR was significantly increased in both EE and CET groups [46]. Furthermore, hippocampal BDNF and VEGF (vascular endothelial growth factor) were also increased by EE and CET. The serum corticosterone (CORT) levels were decreased in EE rats; however, the serum insulin-like growth factor-1 (IGF-1) was increased only in CET rats [46]. 

BDNF has pivotal roles in neurite outgrowth and the expression of synaptic molecules, which are essential for pre- and/or post-synaptic functions in CNS neurons. Therefore, it is probable that GCs affect BDNF-dependent neuronal function. Previously, we reported that DEX exposure inhibited the BDNF-dependent neurite outgrowth and expression of synaptic proteins in cultured hippocampal neurons [47]. In the system, although ERK signaling is important for the action of BDNF-dependent neurite outgrowth and synaptic protein upregulation, DEX decreased the action of the BDNF by repressing ERK signaling [47]. Recently, Restelli et al. (2023) have shown the possible role of Sprouty4 in the negative impact of GCs on the BDNF function [48]. Using PC12 cells, they showed the promotion of transcription of Sprouty4, which is a regulatory molecule repressing ERK signaling stimulated by NGF, after DEX treatment. Importantly, the knockdown of Sprouty4 or an induction of dominant negative Sprouty4 (Y53A) reversed the DEX-dependent inhibition in the NGF/TrkA-promoted ERK activation [48]. They also found that the expression of Sprouty4 was upregulated by DEX in hippocampal neurons, and the knockdown of hippocampal Sprouty4 attenuated the negative influence of DEX on neurite formation and ERK activation stimulated by BDNF, suggesting that Sprouty4 is involved in the inhibitory effects of GCs in neurotrophin function [48]. It was also reported that retinoic acid (RA) signaling through retinoic acid receptor α (RARα) is involved in the regulation of the HPA axis [49,50]. Ke et al. (2019) examined the possible effects of Ro41-5253, a selective RARα antagonist, using a depression animal model [49]. They found that Ro41-5253 treatment increased sucrose preference in the sucrose preference test (SPT), numbers of crossing and rearing in the open field test (OFT), and reduced the immobility time in the forced swimming test (FST) in rats with depression. Furthermore, the antagonist reduced the serum levels of CORT and increased BDNF, PSD95, synaptophysin (SYP), and MAP2 in the hippocampus of rats with depression, suggesting that the antidepressant-like effects of Ro41-5253 in depressed rats improve the hyperactivity of the HPA axis and hippocampal neuronal dysfunction [49].

Because growing evidence has also demonstrated that the Val66Met variation in human BDNF, where a valine (Val) amino acid is replaced with a methionine (Met) amino acid at position 66 of the BDNF protein, is associated with susceptibility to mental disorders, including depression [51], differences in the neuronal function due to the Val66Met variation are very interesting. Previous reports suggest that the Val66Met variation of BDNF results in an alteration of activity-dependent secretion and intracellular trafficking of the neurotrophin [52,53]. Recently, using animal models, changes in responses to acute stress have also been shown. Musazzi et al. (2022) exposed adult male BDNFVal/Met and BDNFVal/Val knock-in mice to acute restraint stress for 30 min and investigated the levels of both BDNF and CORT [54]. They found that the presynaptic release of glutamate, phosphorylation of cyclic AMP response element-binding protein (CREB), and levels of c-fos (an immediate early gene) in the hippocampus of BDNFVal/Met were higher than those in BDNF Val/Val animals, although the plasma CORT concentration, nuclear GR expression, and its phosphorylation in both BDNFVal/Met and BDNFVal/Val mice were similar [54]. Another recent report has suggested that adolescent GC stress leads to sex-specific disruptions to the fear extinction and the GABAergic system due to the Val66Met genotype of BDNF [55]. During late adolescence, both male and female Val/Val and Met/Met mice received CORT via their drinking water. It was revealed that the CORT exposure selectively abolished the fear extinction of female Met/Met animals, while any effect of sex, genotype, or treatment was not observed for the recovery of fear [55]. When examining three types of interneurons, including parvalbumin, somatostatin, and calretinin, in the amygdala nuclei, changed cell densities of the somatostatin-positive population in female (but not male) animals by the Val66Met genotype and altered calretinin-positive cell densities in female (but not male) animals via CORT treatment were observed, although the parvalbumin-positive cell density was not changed by CORT treatment or the genotype [55].

In glutamatergic neuronal function, the actions of receptors in the postsynaptic sites are important, and the contribution of BDNF and GCs to receptor regulation is an interesting issue. Recently, plasticity-related receptor expression and phosphorylation at the synaptic surface after exposure to CORT and BDNF have been reported [56]. Slices from the sensorimotor cortex were exposed to BDNF, CORT, or the simultaneous application of both BDNF and CORT (BDNF + CORT) for 30 min, and immunoblotting for NMDA-type or AMPA-type receptors was performed. Although CORT application increased NMDA-type (GluN2A, B) and AMPA-type (GluA1) receptor phosphorylation, BDNF preferentially upregulated the surface levels of these glutamate receptor subunits. Interestingly, BDNF and CORT induced the phosphorylation of receptors and failed to change the synaptic surface expression levels of these receptor subunits [56]. 

Gong et al. (2019) reported a proteomic profiling of the neural cells using a depressive model caused by CORT exposure. Their proteomic profiles of mouse neuronal C17.2 stem cells in vitro and the brains obtained from the depressive-like mice caused by CORT displayed a downregulated expression of mitochondrial oxidative phosphorylation-related proteins by CORT in addition to reduced BDNF and GR, which were reversed by treatment with berberine (a chemical found in some plants, including coptidis rhizome [57]. Lu et al. (2021) reported a downregulation of Nr3c1 (encoding GR) by cytoplasmic polyadenylation element-binding protein 3 (CPEB3) using a genome-wide screening of CPEB3-bound transcripts [58]. Interestingly, they also found that traumatic intensity-dependent PTSD-like fear memories occurred in CPEB3-KO mice and that hippocampal BDNF downregulation was associated with increased levels of GR during fear extinction in the CPEB3-KO animals, suggesting a possible role of GR-BDNF signaling in fear extinction [58]. 

As shown above, a growing body of evidence indeed demonstrates a close relationship between BDNF dysfunction and chronic stress. An approach concerning the influence of an acute stressor on the BDNF levels in humans is very interesting because the amount of cortisol concentrated under acute physical stress [59] and the possibility of neuronal activity regulation by GCs [60] were reported. Hermann et al. (2021) examined the serum levels of BDNF of healthy young males who received the Trier Social Stress Test (TSST), as there is a possibility that an effect of acute psychosocial stress is dependent on the species [61]. The acute stress significantly increased the serum BDNF concentration in addition to increased cortisol levels compared with the control [61]. It is possible that acute or moderate stressful conditions are suitable for an increased function of the brain as BDNF is upregulated, although the potential mechanisms under transition from the upregulation to the downregulation in BDNF expression levels when receiving stressor stimulation at several levels should be revealed.

Using a meta-analysis and systematic reviews, we accessed the available literature in which BDNF and cortisol analyses were performed; the possible involvement of the Val66Met polymorphism of BDNF in the interplay between BDNF and cortisol contributions in mediating neuronal survival and synaptic plasticity was also examined [62]. Remarkably, the integrated results from the literature demonstrate that BDNF and cortisol play distinct roles, respectively, in the physiology of the brain and that their physiological actions are integrated using GR dynamics. More importantly, the Val66Met polymorphism of BDNF seems to affect individual cortisol responsivity to stress [62], which strongly suggest a critical contribution of the BDNF system and GR signaling in neuronal functions. In the future, a precise understanding of the intracellular interaction between BDNF and GR signaling is needed to further clarify biological systems on the neurotrophin-dependent synaptic plasticity, and more importantly, this will point toward new therapeutic targets for neurodegenerative and psychiatric diseases.

## 5. The Interplay of BDNF and Glucocorticoids in AD

### 5.1. The Role of BDNF in AD

AD is a multifactorial neurodegenerative disorder characterized by progressive cognitive decline, synaptic dysfunction, and memory impairment. Although the exact cause of AD is not fully understood, the amyloid hypothesis is widely recognized as a theory that provides a framework for understanding the pathogenesis of the disease [63]. It proposes that the accumulation of beta-amyloid (Aβ) plaques in the brain is a central event in the development of the disease. Amyloid precursor protein (APP) is a transmembrane protein that is present in many cells, including neurons. In the amyloidogenic pathway, APP is first cleaved by β-secretase (BACE-1), and then the remaining fragment is cleaved by γ-secretase, releasing Aβ peptides of various lengths, including the longer Aβ42 form, which is particularly prone to aggregation. The aggregated Aβ is thought to have neurotoxic effects, leading to abnormal phosphorylation of the tau protein, the subsequent formation of neurofibrillary tangles, the death of neurons, and the progressive cognitive decline observed in AD (Figure 2) [63]. While the disease’s hallmark pathological features are the accumulation of Aβ plaques and neurofibrillary tangles, increasing attention has been directed toward understanding the role of BDNF dysregulation in AD pathophysiology [9,64].

A growing body of evidence suggests that individuals with AD exhibit significant alterations in BDNF levels. Notably, numerous studies have reported a reduction in BDNF expression in the brains of AD patients, particularly in regions susceptible to AD pathology, such as the hippocampus and cortex [65,66]. These changes are indicative of disrupted BDNF homeostasis in AD. In addition to the reduced BDNF in the patients’ brains, the clinical relevance of altered serum BDNF levels in AD becomes apparent when examining their associations with cognitive decline [67]. Lower serum BDNF levels have been consistently correlated with cognitive impairment and the severity of AD symptoms [68,69]. These suggest that BDNF alterations may be an important factor contributing to the clinical manifestations of the disease.

Interestingly, the Val66Met polymorphism in the BDNF gene has been reported to be associated with AD. Recent studies have suggested that individuals carrying the Met allele may have an increased risk of developing AD and may exhibit more severe cognitive decline compared to those with the Val/Val genotype [70,71]. This genetic variation can affect the function and secretion of BDNF [52,72], and it has been associated with alterations in the brain structure and function, as well as various neurological and psychiatric conditions, including AD [51,73].

The mechanisms behind BDNF dysregulation in AD are complex and not fully understood. One prominent factor contributing to decreased BDNF levels is the neurotoxicity associated with Aβ plaques [74]. Aβ oligomers are known to interfere with BDNF signaling pathways and, thus, compromise BDNF availability in the brain [75]. Moreover, mature BDNF levels decreased following intracerebroventricular injection of Aβ1–42 in the rodent hippocampus [76,77]. Additionally, an increase in the proBDNF/mature BDNF ratio was shown after treatment with Aβ25–35 [78,79]. These findings suggest that the proteolytic cleavage of BDNF was also affected by Aβ25–35. Interestingly, fibrillary Aβ25–35 has been shown to selectively elevate the mRNA levels of TrkB-T1, a dominant negative regulator of the full-length TrkB receptor [80].

Neuroinflammation is another key player in the dysregulation of BDNF. In AD, chronic neuroinflammatory processes, characterized by the activation of microglia and astrocytes, lead to the increased production of pro-inflammatory cytokines, such as interleukin-1 (IL-1) [81]. These inflammatory mediators can negatively affect BDNF expression and signaling [82,83]. For example, a study demonstrated that repeated intracerebroventricular (i.c.v.) injections of IL-1 caused a decrease in BDNF mRNA expression [84]. IL-1β was also found to compromise the neurotrophic effects provided by BDNF by suppressing PI3-K/Akt signaling [85].

The interaction between BDNF and other molecules central to AD pathogenesis, including tau protein and apolipoprotein E (ApoE), further complicates BDNF dysregulation in the disease. Recently, Barbereau et al. (2020) demonstrated that the expression of the human Tau-P301L mutation in zebrafish neurons results in a decrease in BDNF expression, while it does not have an impact on TrkB expression [86]. Moreover, a significant decrease in the mRNA levels of BDNF in the frontal cortex was observed five days after the administration of aggregated tau protein into the fourth lateral ventricle of C57Bl/6J mice [87]. 

The APOE gene is the most significant genetic risk factor for late-onset AD, which is the most common form of the disease [88]. Having the ApoE4 allele is associated with an increased risk of developing AD, while having the ApoE2 allele is associated with a decreased risk. ApoE3 is considered neutral in terms of AD risk [89]. Interestingly, ApoE4 has been shown to enhance the nuclear translocation of histone deacetylases (HDACs) in human neurons, leading to a reduction in BDNF gene transcription, while ApoE3 increases histone 3 acetylation and upregulates the expression of BDNF [90]. Moreover, the presence of both ApoE4 and BDNF Val66Met polymorphisms is correlated with a more severe impairment in egocentric navigation and greater atrophy in the medial temporal lobe regions among individuals with amnestic mild cognitive impairment (aMCI) [91]. Pietzuch et al. (2021) also suggested that the interactions between APOE and BDNF polymorphisms may have an impact on the maintenance of functional brain connections in older adults and could potentially represent a vulnerable phenotype associated with the progression of AD [92].

Understanding these molecular interactions is crucial for unraveling the complex relationship between BDNF and AD pathology. It is worth noting that these insights do not imply a direct causative link but rather a complex interplay between multiple factors.

### 5.2. The Role of Glucocorticoids in AD

The role of glucocorticoids in the pathophysiology of AD is also a subject of growing interest and significance. The central concept for understanding the molecular relationship between glucocorticoids and AD is the presence of GRs in the brain. These receptors, primarily located in the hippocampus and prefrontal cortex, are known to mediate the effects of GCs on various physiological processes, including metabolism, immune response, and neural plasticity [93,94].

Recent research has demonstrated that the expression of GRs in key brain regions is linked to vulnerability to AD pathology [95]. The interaction between GCs and these receptors is complex. Under normal physiological conditions, GCs play a crucial role in synaptic plasticity, cognitive function, and memory consolidation [96]. However, chronic exposure to elevated GC levels, a common consequence of chronic stress, can lead to GR dysregulation [97]. This dysregulation may result in a hyperactive stress response and potentially contribute to AD pathogenesis [97]. Epidemiological and clinical investigations have supplied evidence that supports the correlation between chronic exposure to GCs and an elevated risk of developing AD. For example, prolonged exposure to GCs, observed in clinical conditions such as Cushing’s syndrome or extended use of corticosteroid medications, has been linked to cognitive impairment resembling the characteristics of AD [98,99]. Moreover, Zheng et al. (2020) recently demonstrated that AD patients exhibited higher morning cortisol levels compared to controls, and elevated cortisol levels were correlated with accelerated cognitive decline in individuals with mild cognitive impairment (MCI) [100].

Aβ plaques and hyperphosphorylated tau tangles are the hallmarks of AD pathology. Studies have unveiled intriguing links between GCs and the accumulation of these toxic proteins [101,102]. Notably, GCs appear to influence the levels of both Aβ and tau through distinct mechanisms. Research indicates that GCs can promote the production of Aβ peptides, particularly the more aggregation-prone Aβ42 isoform [103]. This effect may be mediated through the modulation of enzymes involved in Aβ production, such as β-secretase [104,105]. Furthermore, GCs have been implicated in impairing the clearance of Aβ from the brain, potentially by interfering with several Aβ-degrading proteases, such as insulin-degrading enzymes and matrix metalloproteinase-9 [106].

In the context of tau pathology, GCs have been associated with the hyperphosphorylation of tau protein [102]. Chronic GC exposure is known to activate kinases responsible for tau phosphorylation, including GSK3, CDK5, and ERK1/2, leading to the formation of neurofibrillary tangles [107,108,109,110]. This tau pathology is closely tied to synaptic dysfunction and neurodegeneration in AD [102].

Neuroinflammation is another crucial element in understanding the role of GCs in AD. Microglia, the resident immune cells of the CNS, play a pivotal role in maintaining brain homeostasis and responding to pathological insults. Recent research has revealed a multifaceted connection between GCs and microglia activation [95,111]. Chronic exposure to elevated GC levels, as seen in conditions of chronic stress, may result in an overactivation of microglia [112]. This hyperactivity can lead to a pro-inflammatory state, characterized by the release of pro-inflammatory cytokines, reactive oxygen species (ROS), and other neurotoxic molecules [112]. Such neuroinflammatory responses are detrimental to neuronal health and have been closely associated with the progression of AD [112]. Additionally, GCs can modulate the microglial phenotype. Under normal conditions, microglia can exhibit both pro-inflammatory (M1) and anti-inflammatory (M2) phenotypes, depending on the context [113]. Dysregulated GC signaling has been shown to skew microglia toward the pro-inflammatory state, exacerbating neuroinflammation in AD [113]. Moreover, microglia-mediated degradation of Aβ, which is crucial for Aβ clearance, may be impaired under conditions of GC dysregulation [114]. 

Oxidative stress is another prominent feature of neuroinflammation and the pathophysiology of AD. GCs have been implicated in modulating oxidative stress pathways, further linking them to neuroinflammatory processes [115]. Elevated GC levels can promote the generation of ROS and reduce the brain’s antioxidant defenses [116]. This imbalance can lead to oxidative damage to cells, proteins, and lipids [117]. Oxidative stress not only directly contributes to neurodegeneration but also exacerbates neuroinflammation by activating microglia and promoting the release of pro-inflammatory mediators [118]. Furthermore, the oxidative damage inflicted by GC-induced oxidative stress may also play a role in the formation of Aβ plaques and tau tangles [119]. Oxidatively modified proteins are more prone to aggregation, and they may contribute to the seeding and propagation of Aβ and tau pathology [119].

Understanding these cellular mechanisms is vital to appreciating the intricate relationship between GC dysregulation and neuroinflammation in AD. Dysregulated GC signaling can create a microenvironment that favors neuroinflammation and oxidative stress, both of which are detrimental to neuronal health. However, it is essential to recognize that while GCs are a piece of this puzzle, they do not act in isolation. Genetic factors, other environmental stressors, and the interplay between different pathological processes in AD must also be considered in the complex pathophysiology of this disease.

### 5.3. BDNF and GCs as Therapeutic Targets in AD

Given the pivotal role of BDNF in neuronal survival, synaptic plasticity, and cognitive function, BDNF dysregulation presents an attractive target for therapeutic interventions in AD. Strategies aimed at restoring BDNF levels and function, either by promoting its production or enhancing its signaling, hold promise for mitigating the cognitive deficits associated with AD.

Several experimental approaches, such as BDNF mimetics [120,121,122,123,124,125], gene therapy [126,127], and lifestyle interventions like physical activity [128,129,130], are being explored as potential ways to address BDNF dysregulation and its consequences (Figure 2). BDNF mimetics are compounds that mimic the actions of BDNF or enhance its receptor binding, promoting neuroprotection and synaptic plasticity. One of these BDNF-enhancing compounds is 7,8-dihydroxyflavone (7,8-DHF), a selective TrkB agonist, as demonstrated by Jang et al. (2010) [131]. Additionally, Yuk-Gunja-Tang (YG), a Korean traditional medicine, has the capacity to enhance the endogenous expression of BDNF [123]. Numerous preclinical studies have provided evidence of the effectiveness of these agents in animal models of AD [120,121,122,123,124,125]. 

Importantly, patients with AD are often prescribed antidepressants, including selective serotonin reuptake inhibitors (SSRIs), to alleviate the depressive symptoms of AD [132]. At the same time, it is well known that SSRIs exert their effects by acting on monoamine transporters, which subsequently results in the activation of BDNF/TrkB signaling [133]. Interestingly, Casarotto et al. (2021) recently identified that SSRIs also directly bind to TrkB receptors, inducing an allosteric potentiation of TrkB signaling [134]. These findings provide support for the notion that BDNF/TrkB signaling serves as the direct target for antidepressant drugs, playing a role in mediating their therapeutic effects in AD patients.

In addition to the pharmacological approaches, lifestyle interventions such as physical exercise and cognitive stimulation have been shown to boost BDNF levels in the brain [130]. A meta-analysis of randomized controlled trials, as reported by Jia et al. (2019), indicated that exercise interventions were linked to significant enhancements in global cognitive function among patients with mild-to-moderate AD [135]. Likewise, research has shown that exercise training can enhance BDNF expression and cognitive function and stimulate neuroplasticity in animal models of AD [128]. Another clinical study also demonstrated that healthy older adults exposed to 35-minute sessions of physical exercise, cognitive training, and mindfulness practice increased serum BDNF levels [136]. Moreover, patients with Parkinson’s disease undergoing cognitive stimulation displayed increased serum BDNF levels as compared to the placebo group [137]. Additionally, Gomutbutra et al. (2022) recently showed that even a brief period of mindfulness-based intervention (MBI) can elevate serum BDNF levels and decrease anxiety in healthy, meditation-naïve females in a randomized, crossover clinical trial [138].

Recognizing the influence of GC dysregulation in AD pathophysiology opens up possibilities for novel therapeutic interventions. Strategies that aim to modulate GC activity represent a promising approach (Figure 2). These include pharmacological interventions to normalize cortisol levels or reduce the sensitivity of glucocorticoid receptors. 11β-HSD1 is a pivotal enzyme that is responsible for the intracellular conversion of inactive cortisone into its active form, cortisol, in humans (or 11-dehydrocortisone into corticosterone in rodents). Consequently, inhibiting 11β-HSD1 results in a decrease in cortisol levels in humans (or CORT levels in rodents) [139,140]. Sooy et al. (2010) showed that UE1961, an inhibitor of 11β-HSD1, demonstrated a significant enhancement in spatial memory performance in aged mice [141]. Moreover, the researchers showed that the administration of another inhibitor, UE2316, led to a decrease in Aβ plaques within the cortex of aged Tg2576 mice [142]. This reduction was concurrent with an elevation in insulin-degrading enzyme (one of the Aβ-degrading proteases) levels, which, in turn, resulted in memory improvements [142]. In addition to 11β-HSD1 inhibitors, selective GR modulators (GRMs) are designed to specifically inhibit GR activity in AD. A study demonstrated that treatment with CORT108297, one of the GRMs, led to a reduction in the levels of APP C-terminal fragments in the 3xTg-AD mouse mode; [143]. Another study also showed that mice that were administered CORT108297 via intraperitoneal injection exhibited a complete reversal of memory deficits, as evaluated through the T-maze test [144].

Besides pharmacological approaches, non-pharmacological interventions aimed at fostering resilience to stress and improving cognitive function could potentially offer benefits in the management of AD [145]. Stress management techniques, such as mindfulness-based interventions and cognitive-behavioral therapy, may be explored as preventive measures for individuals at risk of AD [140]. These approaches could reduce stress-related GR fluctuations and potentially delay disease onset [146,147]. Lifestyle modifications, including regular physical exercise, a balanced diet, and adequate sleep, may also have a significant impact on GC regulation [148]. These factors can help maintain a healthy stress response system and mitigate the effects of chronic stress on neuroinflammation, oxidative stress, and Aβ/Tau pathology [147,149].

BDNF and GCs stand as pivotal factors in the pathophysiology of AD. Their roles in cognitive function, synaptic plasticity, and Aβ/Tau pathology underscore their significance in understanding the disease and their potential as therapeutic targets. Investigating the intricate relationship between BDNF and GCs in AD remains an active area of research, offering hope for novel interventions aimed at mitigating cognitive decline and improving the lives of individuals affected by this challenging neurodegenerative disorder.

## 6. Conclusions

AD, characterized by the progressive loss of cognitive function and memory, remains a significant global health challenge, affecting millions of individuals worldwide. Despite decades of research, our understanding of the intricate mechanisms underlying this debilitating condition is far from complete. While the exact cause of AD is not fully understood, there are several known risk factors. Age is the most significant risk factor, and the risk rises with increasing age. A family history of the disease, genetic factors (such as specific mutations in the APP, PSEN1, and PSEN2 genes), and certain lifestyle factors (such as stress) can also influence one’s risk.

Although its etiology remains multifactorial and complex, an emerging body of evidence suggests that neurotrophic factors and their intricate interplay with the stress hormone cortisol (GC) may play a pivotal role in AD pathogenesis. BDNF, a crucial neurotrophin, plays a central role in maintaining and promoting neuronal health, plasticity, and survival. Recent studies have demonstrated a significant reduction in BDNF levels in the brains of AD patients. Moreover, chronic stress and dysregulated GC signaling have been implicated in the progression of AD. Elevated cortisol levels, commonly associated with chronic stress, can lead to neuronal toxicity and impair synaptic plasticity.

With regard to therapeutic implications, focusing on BDNF and GC signaling pathways holds significant promise as a potential strategy for the management of AD. Pharmacological interventions designed to either enhance BDNF levels or reduce GC signaling have demonstrated their potential in both preclinical and clinical studies of AD. Non-pharmacological strategies, including physical exercise, have also shown beneficial impacts on BDNF and GC levels and could potentially serve as complementary treatments for the disease. However, it is worth noting that there are still several challenges and research gaps that need to be addressed. For instance, it is essential to prudently address safety concerns and potential side effects linked to BDNF interventions. Given that BDNF has diverse effects on various cell types within the CNS and peripheral tissues through the activation of multiple signaling pathways, achieving the desired therapeutic outcomes without inducing adverse reactions can be difficult. Furthermore, the clinical translation of BDNF as a therapeutic agent has been constrained by challenges related to its ability to penetrate the blood–brain barrier (BBB). Neurotrophic factors, including BDNF, are macromolecules, which render them unable to cross the BBB when administered peripherally. Therefore, developing a drug delivery system capable of targeting specific areas within the brain is essential for the effective administration of neurotrophic factors and addressing their high molecular weight. Variability in the response to BDNF- or GC-targeting treatment among individuals with AD represents another limitation. Various factors, including genetic variations, disease stage, and comorbidities, can influence the effectiveness of the therapies. It is crucial to explore the complex interactions between BDNF, stress, and other neurobiological pathways associated with the disorder.

In conclusion, this review illuminates the intricate relationship between BDNF and GC dysregulation in the pathophysiology of AD. As the global burden of AD continues to rise, it is imperative that we decipher the molecular intricacies driving this condition. By targeting BDNF and GC dysregulation, we may unlock novel therapies to improve the quality of life for AD patients and potentially alter the course of this devastating disease. Future research in this area promises to provide a deeper understanding of these mechanisms and pave the way for innovative treatment strategies, offering hope to millions affected by AD.

## Figures and Tables

**Figure 1 ijms-25-01596-f001:**
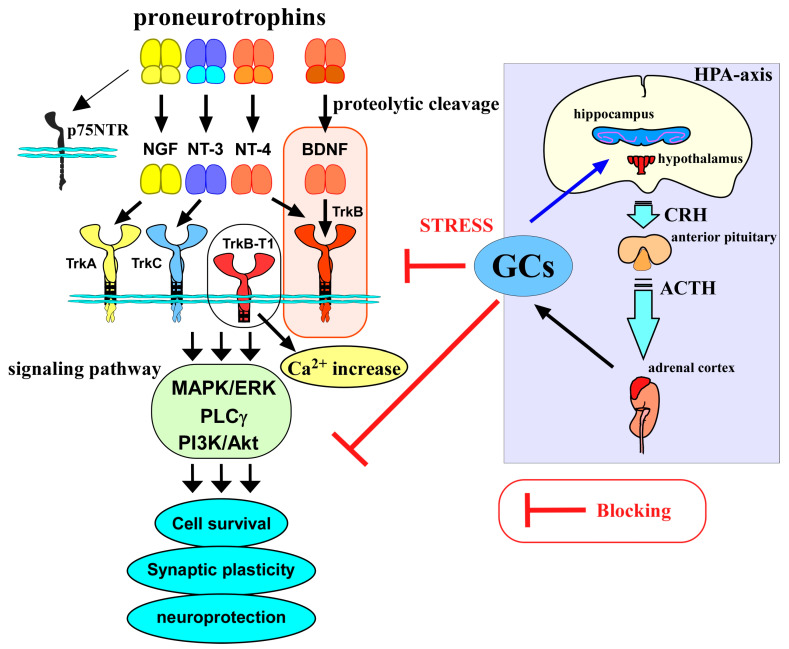
Relationship between the BDNF/TrkB system, glucocorticoids, and HPA axis. Neurotrophins, including nerve growth factor (NGF), BDNF, neurotrophin-3 (NT-3), and neurotrophin-4 (NT-4), are generated as mature forms from the precursor proneurotrophins after receiving proteolytic cleavage. Precursors bind to p75NTR with a high affinity, while mature forms bind to Trks preferentially and contribute to cell survival, regulating synaptic plasticity, and neuroprotection via activating intracellular signaling pathways (MAPK/ERK-, phospholipase Cγ (PLCγ)-, PI3K/Akt pathways).The BDNF/TrkB system is involved in a variety of neuronal events in the CNS. On the other hand, released glucocorticoids (GCs) through activation of the hypothalamic–pituitary –adrenal axis (HPA axis) also have a role in CNS function, and their levels are regulated via a negative feedback action in the HPA axis. GCs are important molecules to cope with stressful conditions; however, abnormally increased levels of GCs under chronic stress exert negative impacts on neurons, including the blockade of the BDNF/TrkB system and/or direct suppression of neuronal functions. CRH, corticotropin-releasing hormone; ACTH, adrenocorticotropic hormone.

**Figure 2 ijms-25-01596-f002:**
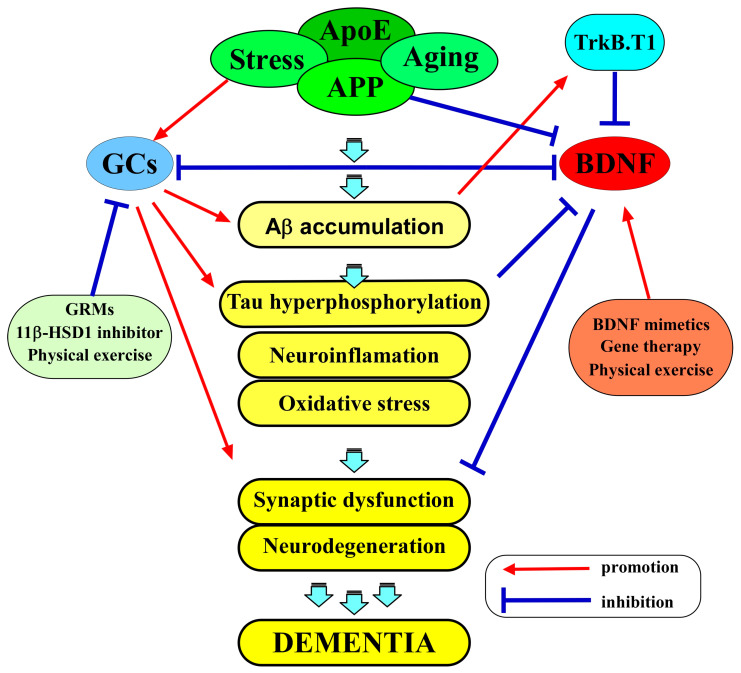
The amyloid hypothesis and the interplay between BDNF and GCs in AD pathogenesis. APP is cleaved by γ-secretase, releasing Aβ peptides in the brain. The aggregated Aβ is thought to have neurotoxic effects, leading to the abnormal phosphorylation of tau protein, the subsequent formation of neurofibrillary tangles, the death of neurons, and the progressive cognitive decline observed in AD. BDNF and GCs exert a multifaceted interplay in the AD brain. BDNF signaling is integral to neuronal survival, synaptic plasticity, and cognitive function. GCs, on the other hand, are central to the body’s response to stress and inflammation. BDNF, through its signaling pathways, may counteract some of the adverse effects of GCs in the brain. BDNF is known to promote neuroprotection, potentially mitigating the neuronal damage caused by excessive GC exposure. Several experimental approaches, such as BDNF mimetics, gene therapy, and lifestyle interventions, like physical activity, are being explored as potential ways to address BDNF dysregulation and its consequences. In addition, strategies that aim to modulate GC activity also represent a promising approach, including GRMs, 11β-HSD1 inhibitors, and physical activity.

## Data Availability

No new data were created or reported in this review article.

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
