# Peer review of "An Interaction between Brain-Derived Neurotrophic Factor and Stress-Related Glucocorticoids in the Pathophysiology of Alzheimer’s Disease"

_ijms, 2024, doi:10.3390/ijms25031596_

Round 1

Reviewer 1 Report

Comments and Suggestions for Authors

Comments to authors.

The review manuscript reviewed and summarized several molecular mechanisms involved in the interaction among BDNF and stress-related glucocorticoids (GCs) in the pathology of Alzheimer’s disease (AD).  

Manuscript provided overview of i) BDNF/TrkB system in neuronal function ii) GCs and neuronal functions, and iii) GC stress, BDNF, and neuronal functions. Review contributes finally to the ongoing elucidation of the relationship between GCs and BDNF in the context of AD. 

In the first part, the authors reported BDNF expression in glial cells. However BDNF in cerebral endothelial cells is not described. The second part concerning GCs and neuronal functions is not easy to read and too complex. I do not understand the sentence page 5 from the line 171 to 176. In the section that follows, I do not understand page 6 from the line 248 to 254. These parts must be revised

 Manuscript is well written and adressed all relevant topics under review with up-to-date literature. There are no objective errors and short-coming worth mentioning. However, Authors need to check fig1 because Trkb-T1 is not involved in BDNF-signaling pathway. Moreover, the function of this receptor needs to be better explained especially in astrocytes. Despite the various parts, the more important description is the interplay of BDNF and GCs in AD and there’s no illustration figure for this part. The pathophysiological context for AD must be detailed. Moreover, the therapeutic approaches targeting BDNF and GCs on AD models must be developed and not listed in a table.

References appear to be too many. Some of the older references may be removed, particularly where other references are available. 

Comments on the Quality of English Language

Moderate editing of english language is required

Author Response

Thank you very much for taking the time to review this manuscript. Please find the detailed responses below and the corresponding revisions/corrections highlighted/ in red in the re-submitted file.

Point 1: In the first part, the authors reported BDNF expression in glial cells. However, BDNF in cerebral endothelial cells is not described.

Response 1: Thank you for your important suggestion. According to the suggestion, we have written about BDNF in cerebral endothelial cells (Page 3, lines 142-152).

Point 2: The second part concerning GCs and neuronal functions is not easy to read and too complex. I do not understand the sentence page 5 from the line 171 to 176. In the section that follows, I do not understand page 6 from the line 248 to 254. These parts must be revised.

Response 2: Thank you for your suggestion. As the reviewer’s suggestion, we have improved the sentences in the sections (Page 5, lines 208-217 and Page 7, lines 286-296).

Point 3: Manuscript is well written and addressed all relevant topics under review with up-to-date literature. There are no objective errors and short-coming worth mentioning. However, Authors need to check fig1 because Trkb-T1 is not involved in BDNF-signaling pathway.

Response 3: Thank you very much for your critical suggestion. As the reviewer’s suggestion, we have added the involvement of Trkb-T1 in BDNF-signaling pathway (Please see Figure 1).

Point 4: Moreover, the function of this receptor needs to be better explained especially in astrocytes.

Response 4: Thank you for your important suggestion. We have written about the function of Trkb-T1 in astrocytes as the reviewer’s suggestion (Page 3, lines 126-135).

Point 5: Despite the various parts, the more important description is the interplay of BDNF and GCs in AD and there’s no illustration figure for this part.

Response 5: We have created figure 2 describing the interplay of BDNF and GCs in AD as the reviewer’s suggestion (Please see Figure 2).

Point 6: The pathophysiological context for AD must be detailed.

Response 6: Thank you for your important suggestion. According to the suggestion, we have written about the pathophysiological context for AD (Page 10, lines 450-461).

Point 7: Moreover, the therapeutic approaches targeting BDNF and GCs on AD models must be developed and not listed in a table.

Response 7: As the reviewer’s suggestion, we have deleted the table and included the therapeutic approaches targeting BDNF and GCs in Figure 2.

Point 8: References appear to be too many. Some of the older references may be removed, particularly where other references are available.

Response 8: According to the suggestion, we have tried to remove some of the old literatures, however, other reviewers’ suggestions indeed required additional references to include new paragraphs in the process of this revision. Further, we believe that the present manuscript form that is informative is suitable for this IJMS journal.   

Point 9: Moderate editing of English language is required.

Response 9: Thank you very much for the critical comments. This revised manuscript has been checked by an English-speaking scientist.

Reviewer 2 Report

Comments and Suggestions for Authors

Considering the growing interest of researchers in the comprehensive assessment of risk factors for Alzheimer's disease, the review by Numakawa and Kajihara is timely and interesting. In general, I think the subject of this article may be of interest to the readers of IJMS. However, some comments needed to be addressed to improve the quality of the manuscript prior to its publication in the present form. My overall judgment is to publish this paper after the authors have carefully considered my suggestions below.

Majors:

1) The text will benefit if the authors provide data from longitudinal studies of the contribution of stress and cortisol in patients with Alzheimer's disease.

2) It seems to me that the authors of the review missed the fact that patients with AD are often prescribed antidepressants. SSRIs appear to be effective in relieving symptoms of depression in patients with AD (Zhang et al., 2023, PMID: 37259037). At the same time, it is well known that SSRIs enhance the expression of BDNF and can also normalize the HPA axis (although this effect varies for individual drugs and depends on the severity of the disease). I think it would be interesting if the authors at least briefly discussed this point in the article.

3) Most of paragraph 5.3 (lines 518-528) generally repeats the content of paragraph 4 and it would be more appropriate there.

4) When talking about TrkB.T1, the authors forgot to mention that this isoform has its own signaling and plays an important role in the sequestration of BDNF, especially in astrocytes (see review Tessarollo and Yanpallewar, 2022, PMID: 35321093). Moreover, a number of studies have shown the contribution of TrkB.T1 to stress vulnerability (Razzoli et al., 2011, PMID: 21272243; Tsai et al., 2014, PMID: 24550802; Barfield et al., 2017, PMID: 29186135; Azogu et al., 2018, PMID: 29753050; Pagliusi et al., 2022, PMID: 35722560).

Minor:

The authors wrote “…a selective TrkB agonist, as demonstrated by Wurzelmann et al. in 2017” (lines 552-553). In fact, the selectivity of 7,8-DHF was demonstrated in the work of Jang et al. in 2010 (PMID: 20133810).

Author Response

Thank you very much for taking the time to review this manuscript. Please find the detailed responses below and the corresponding revisions/corrections highlighted/ in red in the re-submitted file.

Point 1: The text will benefit if the authors provide data from longitudinal studies of the contribution of stress and cortisol in patients with Alzheimer's disease.

Response 1: Thank you for your important suggestion. We have written about longitudinal studies of the contribution of stress and cortisol in patients (Page 11, lines 539-547).

Point 2: It seems to me that the authors of the review missed the fact that patients with AD are often prescribed antidepressants. SSRIs appear to be effective in relieving symptoms of depression in patients with AD (Zhang et al., 2023, PMID: 37259037). At the same time, it is well known that SSRIs enhance the expression of BDNF and can also normalize the HPA axis (although this effect varies for individual drugs and depends on the severity of the disease). I think it would be interesting if the authors at least briefly discussed this point in the article.

Response 2: Thank you for your critical suggestion above. As the reviewer’s suggestion, we discussed the relationship between antidepressants (SSRIs) and BDNF signaling in AD (Page 13, lines 612-619).

Point 3: Most of paragraph 5.3 (lines 518-528) generally repeats the content of paragraph 4 and it would be more appropriate there.

Response 3: According to the suggestion, we have deleted the paragraph 5.3 to remove the redundant content.

Point 4: When talking about TrkB.T1, the authors forgot to mention that this isoform has its own signaling and plays an important role in the sequestration of BDNF, especially in astrocytes (see review Tessarollo and Yanpallewar, 2022, PMID: 35321093). Moreover, a number of studies have shown the contribution of TrkB.T1 to stress vulnerability (Razzoli et al., 2011, PMID: 21272243; Tsai et al., 2014, PMID: 24550802; Barfield et al., 2017, PMID: 29186135; Azogu et al., 2018, PMID: 29753050; Pagliusi et al., 2022, PMID: 35722560).

Response 4: Thank you for your important suggestion. We have written about the function of TrkB-T1 as the reviewer’s suggestion (Page 7, lines 297-330).

Point 5: The authors wrote “…a selective TrkB agonist, as demonstrated by Wurzelmann et al. in 2017” (lines 552-553). In fact, the selectivity of 7,8-DHF was demonstrated in the work of Jang et al. in 2010 (PMID: 20133810).

Response 5: According to the suggestion, we have changed the reference to the work of Jang et al. in 2010 (PMID: 20133810).

Reviewer 3 Report

Comments and Suggestions for Authors

The manuscript ‘An interaction among neurotrophic factor BDNF and stress-related glucocorticoids in the pathophysiology of Alzheimer's disease‘ present studies concerning possible interaction of BDNF/TrkB system with GCs/receptors function in the CNS neurons and discuss the involvement of the interplay of BDNF and GCs in the pathophysiology of AD. The interaction between BDNF and stress-related glucocorticoids is a complex and bidirectional relationship that contributes to the pathophysiology of Alzheimer's disease. Understanding these interactions may open avenues for therapeutic interventions to slow down or prevent the progression of the disease. The submitted manuscript has partially contributed in this aspect.

After going through the manuscript, I have following comments for the author.

1.      Previous studies have reported negative impact of chronic stress and elevated glucocorticoid levels on  the hippocampus -  a brain region crucial for learning and memory. This has known to result in atrophy of the hippocampus, contributing to cognitive decline. I would suggest elaborating this point in the manuscript with appropriate references.

2.      Lifestyle interventions, such as stress management, physical exercise, and cognitive stimulation, are known to help in maintaining optimal BDNF levels and reducing the impact of stress on the brain. This message is briefly mentioned in the manuscript but not elaborately. Please discuss this point with description of various lifestyle interventions and their effects.

Comments on the Quality of English Language

The manuscript has several grammatical and syntax errors. Please double check the manuscript to correct those errors.

Author Response

Thank you very much for taking the time to review this manuscript. Please find the detailed responses below and the corresponding revisions/corrections highlighted/ in red in the re-submitted file.

Point 1: Previous studies have reported negative impact of chronic stress and elevated glucocorticoid levels on the hippocampus - a brain region crucial for learning and memory. This has known to result in atrophy of the hippocampus, contributing to cognitive decline. I would suggest elaborating this point in the manuscript with appropriate references.

Response 1: Thank you for your suggestion. According to the suggestion, we have written about impact of chronic stress and elevated glucocorticoid levels on the hippocampus (Page 2, lines 49-53).

Point 2: Lifestyle interventions, such as stress management, physical exercise, and cognitive stimulation, are known to help in maintaining optimal BDNF levels and reducing the impact of stress on the brain. This message is briefly mentioned in the manuscript but not elaborately. Please discuss this point with description of various lifestyle interventions and their effects.

Response 2: Thank you for your important suggestion above. As the reviewer’s suggestion, we developed the description of various lifestyle interventions and their effects in maintaining optimal BDNF levels and reducing the impact of stress (Page 13, lines 626-633).

Point 3: The manuscript has several grammatical and syntax errors. Please double check the manuscript to correct those errors.

Response 3: Thank you very much for the critical comments. This revised manuscript has been checked by an English-speaking scientist.

Round 2

Reviewer 1 Report

Comments and Suggestions for Authors

The comments to the authors have been answered and are in accordance with my requests.

Reviewer 2 Report

Comments and Suggestions for Authors

The authors adequately responded to all comments. I am completely satisfied with the answers and believe that in its present form the manuscript can be accepted for publication.